# META-BLACK-BOX-OPTIMIZATION THROUGH OF-FLINE Q-FUNCTION LEARNING WITH MAMBA ARCHITECTURE

## ABSTRACT

Recent progress in Meta-Black-Box-Optimization (MetaBBO) has demonstrated that meta-training a neural network based meta-level control policy over an optimization task distribution could significantly enhance the optimization performance of the low-level black-box optimizers. However, achieving such performance enhancement requires effective policy optimization/search method to locate optimal control policy within a massive joint-action space. The online learning fashion of existing works further makes the efficiency of MetaBBO problematic. To address these technical challenges, we propose an offline learning framework in this paper, termed Q-Mamba. Concretely, our method uses a Mamba neural network architecture to meta-learn decomposed Q-functions for each configurable component in the low-level optimizer. By decomposing the Q-function of the configuration decisions of all components in an optimizer, we can apply effective sequence modelling to avoid searching the control policy in the massive joint-action space. Furthermore, by leveraging the long-sequence modelling advantage of Mamba and moderate offline trajectory samples, Q-Mamba can be efficiently trained through a synergy of offline Temporal-Difference update and Conservative Q-Learning regularization to achieve competitive performance against the online learning paradigms. Through extensive benchmarking, we observe that Q-Mamba achieves competitive or even superior optimization performance to prior online/offline learning baselines, while significantly improving the training efficiency of existing online learning baselines. Additional ablation studies show that each of the proposed key designs contributes to this good performance.

## 1 INTRODUCTION

Optimization is everywhere. When it comes to the Black-Box Optimization (BBO), where neither the problem formulation nor the gradient information is accessible, global optimization algorithms in Evolutionary Computation (EC) show superiority for addressing these through better exploration and exploitation tradeoff (Zhan et al., 2022). For decades, a broad family of evolutionary algorithms and swarm intelligence algorithms have been extensively studied and the corresponding application scenarios range from basic engineering problems (Slowik & Kwasnicka, 2020) to advanced scientific discovery (Chen et al., 2023; Guo et al., 2024b). Despite the good performance observed in various BBO problems, one particular technical bottleneck shared by these BBO optimizers is the generalization across different problems (Eiben & Smit, 2011). Typically, to solve a particular optimization problem, deep expertise is required to configure an existing optimizer or redesign a new one. This impedes the further spread of EC towards wider application range.

Recent research efforts in Meta-Black-Box-Optimization (MetaBBO) address the aforementioned generalization gap by introducing a bi-level learning to optimize paradigm (Ma et al., 2023), where a neural network-based control policy is maintained at the meta level and meta-trained to serve as experts for tuning the low-level BBO optimizers (as shown in the top left of Figure 1). However, achieving such generalization performance through meta-learning comes with certain challenges. On the one hand, controlling/configuring all components within the low-level optimizer requires effective policy optimization paradigm such as Reinforcement Learning (RL) (Sutton, 2018) to search for the optimal control policy in a massive joint-action space. On the other hand, to ensure the effec-

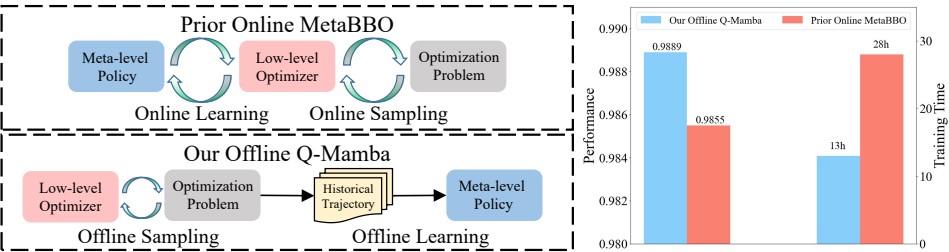

Figure 1: **Top left**: The workflow of existing MetaBBO methods, online learning fashion. **Bottom left**: The workflow of our offline Q-Mamba. **Right**: The normalized performance and training wall time comparison between our offline learning Q-Mamba and online learning MetaBBO.

tiveness of the learning, existing MetaBBO methods primarily facilitate online learning to meta-train their meta-level policies, which is inefficient particularly for the black-box optimization process which typically involves at least hundreds of optimization iterations.

Given a such dilemma in-between the effectiveness and efficiency, we in this paper propose an offline learning MetaBBO framework (as shown in the bottom left of Figure 1), termed **Q-Mamba**, to break the tie and hence ensure both the learning effectiveness and efficiency (as shown in the right of Figure 1). Concretely, to reduce the difficulty of learning optimal control policy from the entire configuration space of a black-box optimizer, we introduce a *Decomposed Q-function Representation* (DQR) which allows sequence modelling-based representation for each component's Q-function. Such decomposition has been studied in some pioneer offline RL researches (Janner et al., 2021; Chebotar et al., 2023) and demonstrated effectiveness in control problems. With the DQR, we further design a *Mamba* (Gu & Dao, 2023) *neural network-based RL agent* (Q-Learner), which treats the configuration of each component in the optimizer as a separate time step and auto-regressively predicts the corresponding decomposed Q-function by conditioning on the current optimization status and configurations of components selected before. To improve the efficiency of training the Q-Learner, we refer to offline RL (Levine et al., 2020), which learns the optimal control policy from demonstration. However, the offline RL training is vulnerable due to the endemic distributional shift issue (Wang et al., 2021) and demonstration quality issue (Ball et al., 2023). To succeed the training, we additionally integrate a *Conservative Q-Learning* (Kumar et al., 2020) regularization (CQL) into the original Bellman backup to relieve the potential distribution shift. Besides, we construct an *Exploration&Exploitation Trajectory Collection* (E&E Dataset) from a mix of randomly generated trajectories and well-performing MetaBBO trajectories. Such a combination enables better offline learning effectiveness. Accordingly, we summarize our contributions in this paper as follows:

- **Novel Framework.** Our main contribution in this paper is Q-Mamba, a novel offline RL MetaBBO framework which shows better learning effectiveness and efficiency than prior online/offline learning baselines.

- **Key Designs.** From the perspective of deep learning, we have defined the *problem formulation* (Section 4.1) as optimizing the Decomposed Q-function (DQR) for each components in a black-box optimizer to reduce the difficulty hence improve the effectiveness of learning from joint-action space, and proposed the *model* (Section 4.2) as a Mamba-based Q-Learner to enhance long-sequence modelling and learning efficiency for configuring the optimizer within the optimization process. To further stabilize the offline training, we have introduced a CQL regularization into the original *training objective* (Section 4.3) and constructed an E&E Dataset as the offline *training data* (Section 4.4), to relieve the distributional shift and reinforce the data quality during the training respectively.

- **Superior Performance.** Experimental results show that our Q-Mamba effectively achieves at least competitive optimization performance against prior online/offline learning baselines, while consuming at most half training budget of the online baselines. The learned meta-level policy can also be readily applied to enhance the performance of the low-level optimizer on unseen BBO tasks, e.g., Neuroevolution (Such et al., 2017) tasks.

## 2 RELATED WORKS

### 2.1 META-BLACK-BOX-OPTIMIZATION

Meta-Black-Box-Optimization (MetaBBO) aims to learning the optimal policy that boosts the optimization performances of the low-level optimizer over a group of optimization problems (Ma et al., 2023). Although several works facilitate supervised learning (Chen et al., 2017; Song et al., 2024; Li et al., 2024b), Neuroevolution (Lange et al., 2023b;a; Ma et al., 2024a) or even LLMs (Ma et al., 2024c; Liu et al., 2024) to meta-learn the control policy, the majority of current MetaBBO methods adopt rather reinforcement learning for the policy optimization to strike a balance between effectiveness and efficiency (Li et al., 2024a). Specifically, the dynamic algorithm configuration during the low-level optimization can be regarded as a Markov Decision Process (MDP), where the state reflects the status of the low-level optimization process, action denotes the configuration space of the low-level optimizer and a reward function is designed to provide feedback to the meta-level control policy. Existing MetaBBO methods differ with each other in the action space. In general, the configuration space of the low-level optimizer involves the operator selection and/or the hyper-parameter tuning. For the operator selection, initial works such as DE-DDQN (Sharma et al., 2019) and DE-DQN (Tan & Li, 2021) facilitate Deep Q-network (DQN) (Mnih, 2013) as the meta-level policy and dynamically suggest one of the prepared mutation operators at each optimization step for the low-level Differential Evolution (DE) (Storn & Price, 1997) optimizer. Following such paradigm, PG-DE (Zhang et al., 2024) and RL-DAS (Guo et al., 2024a) further explore the possibility of using Policy Gradient (PG) (Schulman et al., 2017) methods to train probability model for the operator selection and demonstrate PG methods are more effective than DQN methods. Besides, RLEMMO (Lian et al., 2024) and MRL-MOEA (Wang et al., 2024) extend the target optimization problem domain from single-objective optimization to multi-modal optimization and multi-objective optimization respectively. Unlike the operator selection, the action space in hyper-parameter tuning is not merely discrete since typically the hyper-parameters of an optimizer are continuous with feasible ranges. In such continuous setting, the action space is infinite and can be handled either by discretizing the continuous value range to reduce this space (Liu et al., 2019; Xu & Pi, 2020; Hong et al., 2024; Yu et al., 2024) or directly using PG methods for continuous control (Yin et al., 2021; Sun et al., 2021; Wu & Wang, 2022; Ma et al., 2024b).

While simply doing operator selection or hyper-parameter tuning for part of an optimizer has shown certain performance boost, recent MetaBBO researches such as MADAC (Xue et al., 2022) and ALDes (Zhao et al., 2024) indicate that controlling both sides gains more. However, the massive action space in such setting and the online RL process in these MetaBBO methods make it challenging to balance the training effectiveness and the efficiency. In this paper, we propose Q-Mamba as a novel MetaBBO method to control both the operator selection and hyper-parameter tuning with competitive optimization performance against previous baselines, while reducing training efficiency owing to the proposed sequential Q-function representation and offline learning strategy.

### 2.2 OFFLINE REINFORCEMENT LEARNING

Offline RL (Levine et al., 2020) aims at learning the optimal control policy from a pre-collected demonstration set, without the direct interaction with the environment. This is appealing for real-world complex control tasks, where on-policy data collection is extremely time-consuming (i.e., the dynamic algorithm configuration for black-box optimization discussed in this paper). A critical challenge in offline RL is the distribution shift (Fujimoto et al., 2019): learning from offline data distribution might mislead the policy optimization for out-of-distribution transitions hence degrades the overall performance. Common practices in offline RL to relieve the distribution shift include a) learning policy model (e.g., Q-value function) by sufficiently exploiting the Bellman backups of the transition data in the demonstration set and constraining the value functions for out-of-distribution ones (Haarnoja et al., 2018; Kumar et al., 2020). b) conditional imitation learning (Chen et al., 2021; Janner et al., 2021; Dai et al., 2024) which turns the MDP into sequence modelling problem and uses sequence models (e.g., recurrent neural network, Transformer or Mamba) to imitate state-action-reward sequences in the demonstration data. Although the conditional imitation learning methods have been used successfully in control domain, they do not provide any mechanism to improve the demonstrated behaviour as those policy model learning methods. A recent offline RL method, termed Q-Transformer (Chebotar et al., 2023), combines the strength of both lines of works

by first decomposing the Q-value function for the entire high-dimensional action space into separate one-dimension Q-value functions, and then leveraging transformer architecture for sequential Bellman backups learning. Q-Transformer allows policy improvement during the sequence-to-sequence learning hence achieves superior performance to the prior works. Following Q-Transformer, in this paper, we propose a novel Mamba-based architecture to further enhance the long sequence processing and learning ability under MetaBBO setting.

## 3 PRELIMINARIES

### 3.1 DECOMPOSED Q-FUNCTION REPRESENTATION

Suppose we have an MDP $\{S, A = (A_1, A_2, ...., A_K), R, \mathcal{T}, \gamma\}$, where the action space is associated by a series of $K$ action dimensions, $S$, $R(S, A)$, $\mathcal{T}(S'|S, A)$, $\gamma$ denote the state, reward function, transition dynamic and discount factor, respectively. Value-based RL methods such as Q-learning (Watkins & Dayan, 1992) learn a Q-function $Q(s^t, a_{1:K}^t)$ as the prediction of the accumulated return from the time step $t$ by applying $a_{1:K}^t$ at $s^t$. The Q-function can be iteratively approximated by Bellman backup:

$$Q(a_{1:K}^t|s^t) \leftarrow R(s^t, a_{1:K}^t) + \gamma \max_{a_{1:K}^{t+1}} Q(a_{1:K}^{t+1}|s^{t+1}). \tag{1}$$

However, suppose there are at least $M$ action bins for each of the $K$ action dimensions, the Bellman backup above would be problematic since the associated action space contains $M^K$ feasible actions. Such dimensional curse challenges the learning effectiveness of the value-based RL methods. Recent works such as SDQN (Metz et al., 2017) and Q-Transformer (Chebotar et al., 2023) propose decomposing the associated Q-function into a series of time-dependent Q-function representations for each action dimension to escape the curse of dimensionality. For the $i$-th action dimension, the decomposed Q-function is rewritten as:

$$Q(a_i^t|s^t) \leftarrow \begin{cases} \max_{a_{i+1}^t} Q(a_{i+1}^t|s^t, a_{1:i}^t), & if \quad i < K \\ R(s^t, a_{1:K}^t) + \gamma \max_{a_1^{t+1}} Q(a_1^{t+1}|s^{t+1}). & if \quad i = K \end{cases} \tag{2}$$

Such a decomposition allows using sequence modelling techniques to learn the optimal policy effectively, while holding the learning consistency with the Bellman backup in Eq. (1). We provide a brief proof in Appendix A.

### 3.2 STATE SPACE MODEL AND MAMBA

For an input sequence $x \in \mathbb{R}^{L \times D}$ with time horizon $L$ and $D$-dimensional signal channels at each time step, State Space Model (SSM) (Gu et al., 2022) processes it by the following first-order differential equation, which maps the input signal $x(t) \in \mathbb{R}^D$ to the time-dependent output $y(t) \in \mathbb{R}^D$ through implicit latent state $h(t)$ as follows:

$$h(t) = \overline{A}h(t-1) + \overline{B}x(t), \quad y(t) = Ch(t). \tag{3}$$

Here, $\overline{A}$, $\overline{B}$ and $C$ are learnable parameters, $\overline{A}$ and $\overline{B}$ are obtained by applying zero-order hold (ZOH) discretization rule. An important property of SSM is linear time invariance. That is, the dynamic parameters (e.g., $\overline{A}$, $\overline{B}$ and $C$) are fixed for all time steps. Such models hold limitations for sequence modelling problem where the dynamic is time-dependent. To address this bottleneck, Mamba (Gu & Dao, 2023) lets the parameters $\overline{B}$ and $C$ be functions of the input $x(t)$. Therefore, the system now supports time-varying sequence modelling. In the rest of this paper, we use mamba_block() to denote a Mamba computation block described in Eq. (3).

## 4 Q-MAMBA

In this section, we introduce Q-Mamba, an offline learning-based MetaBBO framework, which enables effective control policy search for black-box optimizers with massive configuration space, through efficient offline reinforcement learning. First, we describe the definition of the settings and formulation of MetaBBO tasks. Next, we elaborate how we apply Q-function decomposition and customized Q-Mamba neural network for sequence modelling of a MetaBBO task. Lastly, we derive the training objective of Q-Mamba and introduce how we collect the offline data for the training.

## 4.1 PROBLEM FORMULATION

A MetaBBO task typically involves three key ingredients: a neural network-based meta-level policy $\pi_\theta$, a black-box optimizer $A$ and a BBO problem distribution $P$ to be solved.

**Optimizer $A$.** Black-box optimizers such as Evolutionary Algorithms (EAs) have been discussed and developed over decades. Initial EAs such as Differential Evolution (DE) (Storn & Price, 1997) holds few hyper-parameters (only two, $F$ and $Cr$ for balancing the mutation and crossover strength). Modern variants of DE integrate various algorithmic components to enhance the optimization performance. Taking the recent winner DE optimizer in *IEEE CEC Numerical Optimization Competition* (Mohamed et al., 2021), MadDE (Biswas et al., 2021) as an example, it has more than ten hyper-parameters, which take either continuous or discrete values. Hence, the configuration space of MadDE is exponentially larger than original DE. In this paper, we use $A : \{A_1, A_2, ..., A_K\}$ to represent an optimizer with $K$ parameters. We use additional $a_i$ to represent the taken value of $A_i$.

**Problem distribution $P$.** By leveraging the generalization advantage of meta-learning, MetaBBO trains $\pi_\theta$ over a problem distribution $P$. A common choice of $P$ in existing MetaBBO works is the *CoCo BBOB Testsuites* (Hansen et al., 2021), which contains 24 basic synthetic functions, each can be extended to numerous problem instances by randomly rotating and shifting the decision variables. Training on all problem instances in $P$ is impractical. We instead sample a collection of $N$ instances $\{f_1, f_2, ..., f_N\}$ from $P$ as the training set. For the $j$-th problem $f_j$, we use $f_j^*$ to represent its optimal objective value, and $f_j(x)$ as the objective value at solution point $x$.

For an optimizer $A$ and a problem instance $f_j$, suppose we have a control policy $\pi_\theta$ at hand and we use $A$ to optimize $f_j$ for $T$ time steps (generations). At the $t$-th generation, we denote the solution population as $X^t$. An optimization state $s^t$ is first computed to reflect the optimization status information of the current solution population $X^t$ and the corresponding objective values $f_j(X^t)$. Then the control policy dictates a desired configuration for $A$: $a_{1:K}^t = \pi_\theta(s^t)$. $A$ optimizes $X^t$ by $a_{1:K}^t$ and obtains an offspring population $X^{t+1}$. A feedback reward $R(s^t, a_{1:K}^t)$ can then be computed as a measurement of the performance improvement between $f_j(X^t)$ and $f_j(X^{t+1})$. The meta-objective of MetaBBO is to search the optimal policy $\pi_{\theta^*}$ that maximizes the expectation of accumulated performance improvement over all problem instances in the training set:

$$\theta^* = \arg\max_\theta \frac{1}{N} \sum_{j=1}^{N} \sum_{t=1}^{T} R(s^t, a_{1:K}^t | \pi_\theta), \quad (4)$$

where such a meta-objective can be regarded as MDP. An effective policy search technique for solving MDP is RL, which is widely adopted in existing MetaBBO methods. In this paper, we focus on a particular type of RL: Q-learning, which performs prediction on the Q-function in a dynamic programming way, as described in Eq. (1).

## 4.2 MAMBA-BASED Q-LEARNER

Existing MetaBBO works primarily struggle in learning meta-level policy with massive joint-action space, which is the configuration space $A : \{A_1, A_2, ..., A_K\}$ associated by $K$ hyper-parameters of the low-level optimizer $A$. To relieve this learning difficulty, we introduce Q-function decomposition strategy as described in Section 3.1. For each hyper-parameter $A_i$ in $A$, we represent its Q-function as a discretized value function $Q_i = \{Q_{i,1}, Q_{i,2}, ..., Q_{i,M}\}$, where $M$ is a pre-defined number of action bins for all $A_i$ in $A$ ($M = 16$ in this paper). For any $A_i$ which takes values from a continuous range, we uniformly discretize the value range into $M$ bins to make universal representation across all $A_i$. By doing this, we turn the MDP in MetaBBO into a sequence prediction problem: we regard predicting each $Q_i$ as a single decision step, then at time step $t$ of the low-level optimization, the complex associated configuration $a_{1:K}^t$ of $A$ can be sequentially decided. We further design a Mamba-based Q-Learner model to assist sequence modelling of decomposed Q-functions. The overall workflow of the Mamba-based Q-Learner is illustrated in Figure 2. We next elaborate technical elements in the figure with their design motivation.

**Optimization state $s^t$.** In MetaBBO, optimization state $s^t$ profiles two types of information: the properties of the optimization problem to be solved and the low-level optimization progress. In Q-Mamba, we construct the optimization state $s^t$ similar with latest MetaBBO methods (Ma et al.,

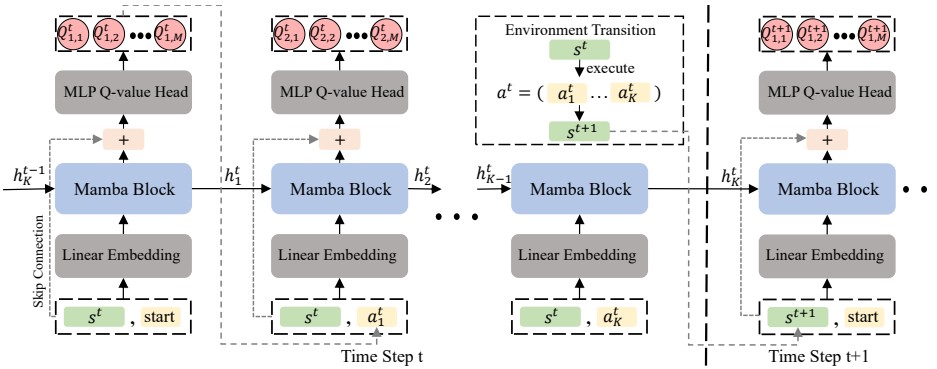

Figure 2: The workflow of the Mamba-based Q-Learner. The forward process of the neural network is similar with the Recurrent Neural Network. At each time step, the Q-function of each decomposed action dimension is output by conditioning the current state and selected action bin of the previous action dimension. The environment transition is executed once all action dimensions are output.

2024b; Chen et al., 2024; Li et al., 2024b). Concretely, at each time step $t$ in the low-level optimization, an optimization state $s^t \in \mathbb{R}^9$ is obtained by calling a function $cal\_state()$. The first 6 dimensions are statistical features about the population distribution, objective value distribution, etc., which provide the problem property information. The last 3 dimensions are temporal features describing the low-level optimization progress. We leave the calculation detail of $s^t$ in Appendix B.

**Tokenization of action bins.** We represent the $M = 16$ action bins of each hyper-parameters $A_i$ in $A$ by 5-bit binary coding: $00000 \sim 01111$. Besides, since we sequentially predict the Q-function for $A_1$ to $A_K$, we additionally use 11111 as a *start* token to activate the sequence prediction. We have to note that for an optimizer $A$, some of its discrete hyper-parameters might hold less than $M$ action bins. For this case, we only use the first several tokens to represent the action bins in these hyper-parameters. In the rest of this paper, we use $token(a_i^t)$ to denote the binary coding of the action bin selected for $A_i$ at time step $t$ of the low-level optimization.

The Mamba-based Q-learner auto-regressively outputs the Q-function values $Q_i^t$ for each $A_i$ in $A$.

**Linear embedding.** To obtain $Q_i^t$, the first step is to prepare the input as the concatenation of the optimization state $s^t$ and the previously selected action bin token $token(a_{i-1}^t)$. Then we apply a linear embedding layer on the input and obtain the embedding feature as follows:

$$\mathbb{E}_i^t = \text{Linear}([s^t, token(a_{i-1}^t)]|W_{emb}, b_{emb}), \tag{5}$$

where $W_{emb} \in \mathbb{R}^{14 \times 16}$ and $b_{emb} \in \mathbb{R}^{16}$ are weights and bias, respectively. For $\mathbb{E}_1^t$, *start* token is used to concat $s^t$, since there is no action bin before $a_1^t$.

**Mamba block.** The computation of the mamba block is described in Section 3.2. It receives the hidden state $h_{i-1}^t$ and the embedding feature $\mathbb{E}_i^t$ and outputs the decision information $\mathbb{O}_i^t$ and hidden state $h_i^t$. $\mathbb{O}_i^t$ is used to parse Q-function $Q_i^t$ and $h_i^t$ is used for next decision step as follows:

$$\mathbb{O}_i^t, h_i^t = \text{mamba\_block}(\mathbb{E}_1^t, h_{i-1}^t|W_{mamba}), \tag{6}$$

where $W_{mamba}$ denotes all learnable parameters in Mamba, which includes the state transition parameters $A$, $B$ and $C$, the parameters of discretization step matrix, and time-varying mapping parameters for the state transition parameters. In this paper we use the mamba-block in Mamba repo[1], with default settings. To obtain $\mathbb{O}_1^t$, the last hidden state of time step $t-1$, $h_K^{t-1}$ is used. The motivation of using Mamba is that: a) For a MetaBBO task, the sequence length involves thousands of decision steps since there are hundreds of optimization steps and $K$ hyper-parameters to be decided per optimization step. We hence adopt Mamba rather than Transformer due to the the inefficiency and performance downside of Transformer for very lone sequence (Ota, 2024), which is addressed by Mamba using data-dependent embedding and hardware-aware design. b) Mamba allows selectively extracting essential information and filter out irrelevant noise according to the input sequence (Gu & Dao, 2023), which would enhance the sequence-to-sequence learning effectively.

---

[1]https://github.com/state-spaces/mamba

**Q-value head.** The Q-value head parses the decision information $\mathbb{O}_i^t$ into the decomposed Q-function $Q_i^t$ through a linear mapping layer. Before the linear mapping, we add the input $[s^t, token(a_{i-1}^t)]$ to $\mathbb{O}_i^t$ as a skip connection as follows:

$$Q_i^t = Norm(\sigma(\mathbb{S}_i^t)), \quad \mathbb{S}_i^t = \text{Linear}(\mathbb{O}_i^t + [s^t, token(a_{i-1}^t)]|W_{head}, b_{head})). \tag{7}$$

Here, $\sigma$ is Leaky ReLU activation function, $Norm$ is the min-max normalization over $M$ bins of $Q_i^t$. $W_{head} \in \mathbb{R}^{16 \times 16}$ and $b_{head} \in \mathbb{R}^{16}$ are weights and bias. When we obtain $Q_i^t$, we select the action bin with the maximum value for hyper-parameter $A_i$: $a_i^t = \arg\max_j Q_{i,j}^t$, and use $token(a_i^t)$ for inferring the decomposed Q-function $Q_{i+1}^t$ of next decision step. Once the action bins of all hyper-parameters $A_1 \sim A_K$ have been decided, the optimizer $A$ optimizes the problem for one step and obtains the optimization state $s^{t+1}$ from the updated solution population. To summarize, in Q-Mamba, the meta-level policy $\pi_\theta$ is the Mamba-based Q-Learner, of which the learnable parameters $\theta$ includes $\{W_{emb}, b_{emb}, W_{mamba}, W_{head}, b_{head}\}$.

### 4.3 TRAINING OBJECTIVE

Online learning is widely adopted in existing works, which is especially inefficient under MetaBBO setting, where the low-level optimization typically involves hundreds of optimization steps hence extremely time-consuming. In this paper we propose learning the decomposed sequential Q-function through offline RL to improve the training efficiency of MetaBBO. Concretely, we consider a trajectory $\tau = \{s^1, (a_1^1, ..., a_K^1), r^1, ..., s^T, (a_1^T, ..., a_K^T), r^T\}$, which is previously sampled by an offline policy $\hat{\pi}$. Here, $a_i^t$ denotes the action bin selected for $A_i$ at time step $t$. The training objective of Q-Mamba is a synergy of Bellman backup update (Eq. (2)) and conservative regularization as

$$J(\tau|\theta) = \sum_{t=1}^{T} \sum_{i=1}^{K} \sum_{j=1}^{M} J(Q_{i,j}^t|\theta) = \begin{cases} \frac{1}{2}(Q_{i,j}^t - \max_j Q_{i+1,j}^t)^2, & if \quad i < K, j = a_i^t \\ \frac{\beta}{2}\left[Q_{i,j}^t - (r^t + \gamma \max_j Q_{1,j}^{t+1})\right]^2, & if \quad i = K, j = a_i^t \\ \frac{\lambda}{2}(Q_{i,j}^t - 0)^2, & if \quad j \neq a_i^t \end{cases} \tag{8}$$

where $Q_{i,j}^t$ is the Q-value of the $j$-th bin in $Q_i^t$, which is output by our Mamba-based Q-Learner $\pi_\theta$, with $[s^t, token(a_{i-1}^t)]$ as input. The first two branches in Eq. (8) are TD error following the Bellman backup for decomposed Q-function (as described in Eq. (2)). We additionally add a weight $\beta$ (we set $\beta = 10$ in this paper) on the last action dimension to reinforce the learning on this dimension. As described in Eq. (2), the other action dimension is updated by the inverse maximization operation, so ensuring the accuracy of the Q-value in the last action dimension helps secure the accuracy of the other dimensions. The last branch in Eq. (8) is the conservative regularization introduced in representative offline RL method CQL (Kumar et al., 2020), which is used to relieve the over-estimation due to the distribution shift. Here, the Q-values of action bins which are not selected in the trajectory $\tau$ ($j \neq a_i^t$) is regularized to 0. This would accelerate the learning of the TD error. We set the weight of the conservative regularization $\lambda = 1$ in this paper.

### 4.4 E&E DATASET

The trajectory samples play a key role in offline RL applications (Ball et al., 2023). On the one hand, good quality data helps the training converges. On the other hand, randomly generated data help RL explores and learns more robust model. In Q-Mamba, we collect a trajectory dataset $\mathbb{C}$ of size $D = 10K$ which combines the good quality data and randomly generated data. Concretely, for a low-level black-box optimizer $A$ with $K$ hyper-parameters and a problem distribution $P$, we pre-train a series of up-to-date MetaBBO methods (e.g., RLPSO (Wu & Wang, 2022), LDE (Sun et al., 2021), GLEET Ma et al. (2024b)) which control hyper-parameters of $A$ to optimize the problems in $P$. Then we rollout the pre-trained MetaBBO methods on problem instances in $P$ to collect $\mu \cdot D$ complete trajectories. We then use the random strategy to randomly control the hyper-parameters of $A$ to optimize the problems in $P$ and collect $(1 - \mu) \cdot D$ trajectories. By combining the exploitation experience in the trajectories of MetaBBO methods and the exploration experience in the random trajectories, our Q-Mamba learns robust and high-performance meta-level policy. In this paper, we set $\mu = 0.5$ to strike a good balance. To meta-train a Q-Mamba agent for controlling $A$ to optimize problems in $P$, we use AdamW with a learning rate $5e - 3$ to minimize the expectation

training objective $\mathbb{E}_{\tau \in \mathbb{C}} J(\tau|\theta)$. The training lasts for 300 epochs with a batch size of 64. After the training, the learned Q-Learner model $\pi_\theta$ can be directly used to control $A$ for unseen problems. These unseen problems can be either those within the same problem distribution $P$, or totally out-of-distribution ones. We validate both generalization aspects of our Q-Mamba in the following experimental section.

## 5 EXPERIMENTAL RESULTS

In the experiments, we aim to answer the following questions: a) How Q-Mamba performs compared with the other online/offline baselines? b) Can Q-Mamba be zero-shot to more challenging realistic optimization scenario? c) How important are the key designs in Q-Mamba?

### 5.1 EXPERIMENT SETUP

**Training dataset.** We have prepared 10 different low-level black-box optimizer $Alg0 \sim Alg9$, which cover several types of algorithms such as DE, PSO and GA. Due to the different algorithm structure inside, the number of hyper-parameters (action dimensions) in these optimizers range from $3 \sim 16$, hence showing different difficulty-levels for MetaBBO methods. We introduce how we construct these optimizer and their algorithm structures in Appendix D.1. The problem distribution selected for the training is the *CoCo BBOB Testsuites* (Hansen et al., 2021), which contains 24 basic synthetic functions with diverse properties such as uni-modal, multi-modal, (non-)separable, (a)symmetrical, flattened areas, and continuity features. We denote it as $P_{bbob}$. We further facilitate train-test split on $P_{bbob}$, dividing it into 16 problem instances for the training, and 8 problem instances for the testing. These problem instances range from $5 \sim 50$-dimensional, we randomly apply shift and rotation on their solution spaces to make the optimization landscapes more challenging. Details of $P_{bbob}$ and its train-test split is provided in Appendix D.2. By using $Alg0 \sim Alg9$ and the 16 training problem instances, we create 10 E&E Datasets by the procedure described in Section 4.4. For online baselines, we train them on each low-level optimizer to optimize the training problem instances. For offline baselines including our Q-Mamba, we train them on each E&E Dataset. We note that the total optimization steps for the low-level optimization is set as $T = 500$.

**Baselines.** We compare a wide range of baselines to obtain comprehensive and significant experimental observations. Concretely, we compare four **online baselines**: RLPSO (Wu & Wang, 2022) that uses simple MLP architecture for controlling low-level optimizers. LDE (Sun et al., 2021) that facilitates LSTM architecture for sequential controlling low-level optimizers using temporal optimization information. GLEET (Ma et al., 2024b) that uses Transformer architecture for mining the exploration-exploitation tradeoff during the low-level optimization. These three baselines are all trained to output associated configuration without decomposition as our Q-Mamba. We also provide an online baseline of our Q-Mamba, which learns by interacting with the environments. We also compare two **offline baselines**: Decision Transformer (Chen et al., 2021) and Q-Transformer (Chebotar et al., 2023). The former tokenizes the state, action and return-to-go signal and uses Transformer for sequence-to-sequence fitting, which is an offline RL method through conditional imitation learning. The latter applies Q-function decomposition as our Q-Mamba and facilitates offline Q-learning. However it has to split the trajectory sequence into short context windows for Transformer to process and hence is claimed relatively weak in super long sequence modelling such as the decomposed Q-value sequence in this paper. The settings of these baselines primarily follows their original papers, with a little fix up to make it compatible with the tasks in this paper. We elaborate them in Appendix D.3. To ensure the fairness of the comparison, all baselines go through the same order of training data, which is $10K$ trajectories.

**Performance metric.** We adopt the accumulated performance improvement $Perf(A, f|\pi_\theta)$ for measuring the optimization performance of the compared baselines and our Q-Mamba. Given a MetaBBO baseline $\pi_\theta$, the corresponding low-level optimizer $A$ and an optimization problem instance $f$, the accumulated performance improvement is calculated as the sum of reward feedback at each optimization step $t$: $Perf(A, f|\pi_\theta) = \sum_{t=1}^{T} r^t$. The reward feedback is calculated as the relative performance improvement between two consecutive optimization steps: $r^t = \frac{f^{*,t-1} - f^{*,t}}{f^{*,0} - f^*}$, where $f^{*,t}$ is the objective value of the best found solution until time step $t$, $f^*$ is the optimum of $f$. The maximal accumulated performance improvement is 1 when the optimum of $f$ is found.

Table 1: Performance comparison between Q-Mamba and other online/offline baselines. All baselines are tested on unseen problem instances within the training distribution $P_{bbob}$. We additionally present the averaged training/inferring time of all baselines in the last row.

| | Online | | | | Offline | | |
|---|---|---|---|---|---|---|---|
| | RLPSO (MLP) | LDE (LSTM) | GLEET (Transformer) | Online Q-Mamba | Decision-Transformer | Q-Transformer | Q-Mamba |
| $Alg0$ | 9.855E-01 ±9.038E-03 | 9.563E-01 ±1.830E-02 | 9.616E-01 ±3.110E-03 | 9.873E-01 ±2.096E-01 | 9.325E-01 ±2.680E-02 | 9.646E-01 ±3.975E-02 | **9.889E-01** ±**7.779E-03** |
| $Alg1$ | 9.833E-01 ±6.924E-03 | 9.597E-01 ±1.882E-02 | 9.793E-01 ±6.555E-03 | 9.719E-01 ±2.841E-02 | 5.699E-01 ±1.054E-01 | **9.847E-01** ±**6.167E-03** | 9.779E-01 ±3.602E-02 |
| $Alg2$ | 9.542E-01 ±4.945E-02 | **9.747E-01** ±**1.748E-02** | 8.913E-01 ±2.192E-02 | 9.347E-01 ±1.050E-01 | 9.297E-01 ±2.899E-02 | 8.290E-01 ±7.413E-02 | 9.325E-01 ±9.763E-02 |
| $Alg3$ | 9.894E-01 ±7.337E-03 | 9.866E-01 ±2.054E-02 | 9.887E-01 ±3.853E-03 | 9.910E-01 ±6.400E-03 | 7.852E-01 ±5.396E-02 | 9.895E-01 ±9.949E-03 | **9.915E-01** ±**1.962E-02** |
| $Alg4$ | 9.953E-01 ±3.322E-03 | 9.877E-01 ±1.118E-02 | 9.938E-01 ±2.834E-03 | 9.951E-01 ±4.103E-03 | 6.764E-01 ±1.193E-01 | 9.951E-01 ±3.487E-03 | **9.963E-01** ±**7.592E-03** |
| $Alg5$ | 9.740E-01 ±2.250E-02 | 9.857E-01 ±8.725E-03 | 9.795E-01 ±1.501E-02 | 9.841E-01 ±9.374E-02 | 7.265E-01 ±1.011E-01 | 9.474E-01 ±2.329E-02 | **9.865E-01** ±**2.508E-02** |
| $Alg6$ | 9.725E-01 ±1.581E-02 | 9.769E-01 ±1.596E-03 | 9.525E-01 ±2.431E-02 | 9.704E-01 ±3.878E-02 | 9.233E-01 ±3.921E-02 | 8.837E-01 ±5.120E-02 | **9.842E-01** ±**3.285E-02** |
| $Alg7$ | 9.450E-01 ±2.050E-02 | **9.735E-01** ±**1.117E-02** | 9.678E-01 ±1.225E-02 | 9.611E-01 ±2.182E-02 | 8.426E-01 ±4.855E-02 | 9.598E-01 ±3.276E-02 | 9.665E-01 ±6.986E-02 |
| $Alg8$ | 9.924E-01 ±4.745E-03 | 9.867E-01 ±9.023E-02 | 9.898E-01 ±5.875E-03 | 9.9294E-01 ±1.421E-02 | 9.734E-01 ±1.463E-02 | 9.509E-01 ±1.903E-02 | **9.933E-01** ±**2.633E-02** |
| $Alg9$ | 9.914E-01 ±4.497E-03 | 9.904E-01 ±6.306E-03 | 9.910E-01 ±5.846E-03 | 9.920E-01 ±9.485E-03 | 8.706E-01 ±3.951E-02 | 9.895E-01 ±6.754E-03 | **9.950E-01** ±**9.981E-03** |
| Avg Time | 28h / 11s | 28h / 12s | 25h / 13s | 63h / 10s | 13h / 10s | 50h / 11s | 13h / 10s |

## 5.2 IN-DISTRIBUTION GENERALIZATION

After the training, we compare the generalization performance of our Q-Mamba and other baselines on the 8 problem instances in $P_{bbob}$ which are not used for the training of all baselines. Concretely, for each baseline and each low-level optimizer, we report in Table 1 the average value and error bar of the accumulated performance improvement across the 8 tested problems and 19 independent runs. We additionally present the average training time and inferring time (time consumed to complete a trajectory) for each baseline in the last row. The results in Table 1 show that: a) **Q-Mamba v.s. Online baselines.** Q-Mamba significantly outperforms the online baselines RLPSO, LDE and GLEET, which control the low-level optimizer in the massive associated configuration spaces. This evidences the effectiveness of using the decomposed Q-function representation, which could significantly reduce the configuration hence eases the learning difficulty. Meanwhile, due to the offline learning paradigm, Q-Mamba consumes only half of the training time the online baselines require. This is especially appealing for BBO scenarios where the simulation is expensive and time-consuming. b) **Q-Mamba v.s. Decision Transformer.** We observe that Decision-Transformer holds similar training efficiency with our Q-Mamba. The difference between it and Q-Mamba is that DT generally imitates the trajectory by predicting the tokens in the transitions. Results in the table show the performance of DT is quite unstable. In opposite, our Q-Mamba allows policy improvement during the sequence learning, which shows better learning convergence and effectiveness than the conditional imitation-learning based offline RL such as DT. c) **Q-Mamba v.s. Q-Transformer.** While our Q-Mamba shares the Q-function decomposition as a core design, a major novelty we introduced is the Mamba architecture and the corresponding weighted Q-function representation learning. The superior performance of Q-Mamba to the Q-Transformer possibly roots from the inability of Transformer architecture for extremely long Q-function sequence in MetaBBO setting. In Q-transformer, the entire sequence is divided into numerous context windows and learned respectively. Such forced truncation not only influences the long-term temporal dependency but also increases the training time. d) **Q-Mamba v.s. Online Q-Mamba.** We observe a performance degradation when training Q-Mamba under the online learning setting. It might reveal that the offline data provided by the other policies could enrich the experience of the meta-level policy, while online data sorely comes from the meta-level policy itself. The generalization performance is hence degraded.

## 5.3 OUT-OF-DISTRIBUTION GENERALIZATION

We further validate the generalization performance of Q-Mamba and other baselines on more challenging scenario, e.g., neuroevolution (Such et al., 2017) tasks. In a neuroevolution task, a black-box optimizer is used to evolve a population of neural networks according to their performance on a specific machine learning task, i.e., classification, robotic control (Galván & Mooney, 2021). Specifically, we consider four continuous control tasks in Mujoco (Todorov et al., 2012). We optimize a

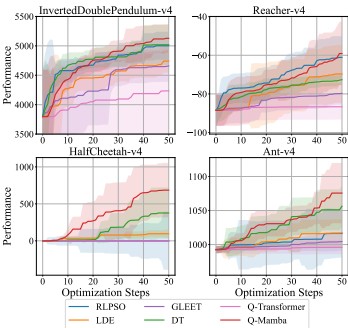

Figure 3: Zero shot performance of Q-Mamba and the other baselines on neuroevolution tasks.

Table 2: Performance analysis on the importance of loss ratio $\lambda$ and $\beta$.

|  | $\lambda = 0$ | $\lambda = 1$ | $\lambda = 10$ |
|---|---|---|---|
| $\beta = 1$ | 9.756E-01 | 9.828E-01 | 9.855E-01 |
|  | $\pm$1.570E-02 | $\pm$1.203E-02 | $\pm$1.192E-02 |
| $\beta = 10$ | 9.833E-01 | **9.889E-01** | 9.857E-01 |
|  | $\pm$1.424E-02 | $\pm$**7.780E-03** | $\pm$1.134E-02 |

Table 3: Performance of Q-Mamba under different proportion of good quality data.

| $\mu$ | 0 | 0.25 | 0.5 | 0.75 | 1 |
|---|---|---|---|---|---|
| Perf. | 9.832E-01 | 9.874E-01 | **9.889E-01** | 9.793E-01 | 9.834E-01 |
|  | $\pm$1.264E-02 | $\pm$6.489E-03 | $\pm$**7.780E-03** | $\pm$1.614E-02 | $\pm$9.692E-03 |

2-layer MLP policy for each task by Q-Mamba and other baselines trained for controlling $Alg0$ on $P_{bbob}$. To align with the challenging condition in realistic BBO tasks, we only allow the low-level optimization involves a small network population (10 solutions) and $T = 50$ optimization steps. We present the average optimization curves across 10 independent runs in Figure 3. The results underscore the superior generalization performance of Q-Mamba to all other baselines: while only trained on synthetic problems with at most 50 dimensions, our Q-Mamba is capable of optimizing the MLP polices which hold thousands of parameters in these neuroevolution tasks.

## 5.4 ABLATION STUDY

We perform two ablation experiments on our Q-Mamba to validate the effectiveness of the key designs. First, we demonstrate the effectiveness over the proposed training objective in Eq. (8). As shown in Table 2, when $\lambda = 0$, the training objective in Eq. (8) turns into the Bellman backup without conservative regularization. The performance degradation under this setting reveals the importance of the conservative term for relieving the distribution shift caused by offline leaning. When $\beta = 1$, the training objective would not focus on the Q-value prediction of the last action dimension, which in turn interferes the prediction of other action dimensions through the inverse maximization operation in Eq. (2). A setting with $\lambda = 1$ and $\beta = 10$ ensures the overall learning effectiveness. Next, we analyse the data mixing ratio $\mu$ in the E&E dataset (Section 4.4). When $\mu = 0$, all trajectories come from a random configuration strategy. When $\mu = 1$, all trajectories come from the well-performing MetaBBO baselines. The results in Table 3 reveal that mixing these two types of data equally ($\mu = 0.5$) might enhance Q-Mamba's learning effectiveness by leveraging the rich historical experiences from both exploration and exploitation.

## 6 CONCLUSION

In this paper, we propose Q-Mamba as a novel offline learning-based MetaBBO framework which improves both the effectiveness and the training efficiency of existing online leaning-based MetaBBO methods. To achieve this, Q-Mamba decomposes the associated Q-function for the massive configuration space into sequential Q-functions for each configuration. We further propose a Mamba-based Q-Learner for effective sequence learning tailored for such Q-function decomposition mechanism. By incorporating with a large scale offline dataset which includes both the exploration and exploitation trajectories, Q-Mamba consumes less than half training time of existing online baselines, while achieving strong control power across various black-box optimizers and diverse BBO problems. Our framework does have certain limitations. First the number of the action bins $M$ cannot be too large under the Q-learning paradigm, this might become cumbersome if fine-grained control is required for some optimizers. Second, Q-Mamba is trained for a given optimizer and requires re-training for other optimizers. An effective optimizer feature extraction mechanism may enhance Q-Mamba's co-training on various optimizers. We mark this as an important future work.

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

## A  PROOF OF Q-FUNCTION DECOMPOSITION

To show that transforming MDP into a per-action-dimension form still ensures optimization of the original MDP, we show that optimizing the Q-function for each action dimension is equivalent to optimizing the Q-function for the full action. We omit the time step superscript $t$ for the ease of presentation.

If we consider apply full action $a_{1:K}$ at the current state $s$ to transit to the next step state $s'$. The Bellman update of the optimal Q-function could be written as:

$$\max_{a_{1:K}} Q(a_{1:k}|s) = \max_{a_{1:K}} \left[ R(s, a_{1:K}) + \gamma \max_{a_{1:K}} Q(a_{1:K}|s') \right]$$
$$= R(s, a_{1:K}^*) + \gamma \max_{a_{1:K}} Q(a_{1:K}|s') \tag{9}$$

where $R(\cdot)$ is the reward we get after executing the full action $a_{1:K}$. Under the Q-function decompostion, the Bellman update of the optimal Q-function for each action dimension $a_i$ is:

$$\max_{a_i} Q(a_i|s, a_{1:i-1}^*) = \max_{a_i} \left[ \max_{a_{i+1}} Q(a_{i+1}|s, a_{1:i}^*) \right]$$
$$= \max_{a_i} \left[ \max_{a_{i+1}} \left( \max_{a_{i+2}} Q(a_{i+2}|s, a_{1:i+1}^*) \right) \right]$$
$$= \cdots$$
$$= R(s, a_{1:K}^*) + \gamma \max_{a_1} Q(a_1|s')$$
$$= R(s, a_{1:K}^*) + \gamma \max_{a_1} \left[ \max_{a_2} Q(a_2|s', a_1) \right]$$
$$= \cdots$$
$$= R(s, a_{1:K}^*) + \gamma \max_{a_{1:K}} Q(a_{1:K}|s') \tag{10}$$

Here the first two lines are the inverse maximization operation as described in Section 3.1, the fourth line is the Bellman update for the last action dimension. The last three lines also follow the inverse maximization operation. By comparing Eq. (9) and Eq. (10) we prove that optimizing the decomposed Q-function consistently optimizes the original full MDP.

## B  OPTIMIZATION STATE DESIGN

The formulation of the optimization state features is described in Table 4. States $s_{\{1\sim6\}}$ are optimization problem property features which collectively represent the distributional features and the statistics of the objective values of the current candidate population. Specifically, state $s_1$ represents the average distance between each pair of candidate solutions, indicating the overall dispersion level. State $s_2$ represents the average distance between the best candidate solution in the current population and the remaining solutions, providing insights into the convergence situation. State $s_3$ represents the average distance between the best solution found so far and the remaining solutions, indicating the exploration-exploitation stage. State $s_4$ represents the average difference between the best objective value found in the current population and the remaining solutions, and $s_5$ represents the average difference when compared with the best objective value found so far. State $s_6$ represents the standard deviation of the objective values of the current candidates. Then, states $s_{\{7,8,9\}}$ collectively represent the time-stamp features of the current optimization progress. Among them, state $s_7$ denotes the current process, which can inform the framework about when to adopt appropriate strategies. States $s_8$ and $s_9$ are measures for the stagnation situation.

## C  ACTION DISCRETIZATION AND RECONSTRUCTION

Given the $M = 16$ bins of Q values $Q_i^t$ for the $i$-th action, if the $i$-th hyper-parameter $A_i$ of the low-level optimizer is in continuous space, we first uniformly discretize the space into $M$ bins: $\hat{A}_i = \{A_{i,1}, A_{i,2}, \cdots, A_{i,M}\}$ where $A_{i,1}$ and $A_{i,M}$ are the lower and upper bounds of the space.

Table 4: Formulations of state features.

| | States | | Notes |
|---|---|---|---|
| **Problem Property** | $s_1^t$ | $\displaystyle \operatorname*{mean}_{x_i, x_j \in X^t} ||x_i - x_j||_2$ | Average distance between any pair of individuals in current population. |
| | $s_2^t$ | $\displaystyle \operatorname*{mean}_{x_i \in X^t} ||x_i - x^{*,t}||_2$ | Average distance between each individual and the best individual in $t$-th generation. |
| | $s_3^t$ | $\displaystyle \operatorname*{mean}_{x_i \in X^t} ||x_i - x^*||_2$ | Average distance between each individual and the best-so-far solution. |
| | $s_4^t$ | $\displaystyle \operatorname*{mean}_{x_i \in X^t} (f(x_i) - f(x^*))$ | Average objective value gap between each individual and the best-so-far solution. |
| | $s_5^t$ | $\displaystyle \operatorname*{mean}_{x_i \in X^t} (f(x_i) - f(x^{*,t}))$ | Average objective value gap between each individual and the best individual in $t$-th generation. |
| | $s_6^t$ | $\displaystyle \operatorname*{std}_{x_i \in X^t} (f(x_i))$ | Standard deviation of the objective values of population in $t$-th generation, a value equals 0 denotes converged. |
| **Optimization Progress** | $s_7^t$ | $(T - t)/T$ | The potion of remaining generations, $T$ denotes maximum generations for one run. |
| | $s_8^t$ | $st/T$ | $st$ denotes how many generations the optimizer stagnates improving. |
| | $s_9^t$ | $\begin{cases} 1 & \text{if } f(x^{*,t}) < f(x^*) \\ 0 & \text{otherwise} \end{cases}$ | Whether the optimizer finds better individual than the best-so-far solution. |

Then we use the action $a_i^t$ obtained by $a_i^t = \arg\max_j Q_{i,j}^t$ as an index and assign the value of the $i$-th hyper-parameter $A_i$ with $A_i = \hat{A}_i[a_i^t]$. If the hyper-parameter is in discrete space $\hat{A}$ with $m_i \leq M$ candidate choices, the action $a_i^t$ is obtained by $a_i^t = \arg\max_{j \in [1, m_i]} Q_{i,j}^t$ and the value of the $i$-th hyper-parameter is $\hat{A}[a_i^t]$. After the value of all hyper-parameters are decided, the optimizer $A$ takes a step of optimization with the hyper-parameters and return the next state from the updated population.

# D   EXPERIMENT SETUP

## D.1   BACKEND ALGORITHM GENERALIZATION

In this paper, we randomly construct 10 optimizers with action space dimensions $\{3, 5, 7, 8, 10, 12, 13, 14, 15, 16\}$. To do so, we first collect a optimization operator space containing operators with controllable parameters such as the mutation and crossover operators from DE (Storn & Price, 1997), PSO update rules (Kennedy & Eberhart, 1995), crossover and mutation operators from GA (Holland, 1992). Operators without controllable parameters such as selection and population reduction operators are also included. Then, to get an optimizer with $n$ hyper-parameters, we randomly sample a batch of operators to construct an optimizer, if the total number of controllable parameters in all operators of the optimizer is not match $n$, we eliminate it and resample until the wanted optimizer is constructed. The hyper-parameters of the optimizer such as the initial population sizes are randomly determined. Below we present the structure of $Alg0$ (3 actions) and $Alg9$ (16 actions) as examples.

$Alg0$ (as shown in Algorithm 1) is DE/current-to-rand/1/exponential (Storn & Price, 1997) with Linear Population Size Reduction (LPSR) (Tanabe & Fukunaga, 2014). The mutation operator DE/current-to-rand/1 is formulated as:

$$x_i' = x_i + F1(x_{r1} - x_i) + F2(x_{r2} - x_{r3}) \tag{11}$$

---

**Algorithm 1** Pseudo code of $Alg0$

---

1: **Input**: Optimization problem $f$, optimization horizon $T$, Meta-level agent $\pi$.
2: **Output**: Optimal solution $x^* = \arg\min\limits_{x \in X} f(x)$.
3: Uniformly initialize a population $X_1$ with shape $NP_1 = 100$ and evaluate it with problem $f$;
4: **for** $t = 1$ **to** $T$ **do**
5:     Receive the 3 action values $a_t = \{F1, F2, Cr\}$ from the agent $\pi$;
6:     Generate $X_t'$ by using DE/current-to-rand/1 (Eq. (11)) on $X_t$;
7:     Apply Exponential crossover (Eq. (12)) on $X_t$ and $X_t'$ to get $X_t''$;
8:     Clip the values beyond the search range in $X_t''$;
9:     Calculate $f(X_t'')$;
10:    Compare $f(X_t)$ and $f(X_t'')$, select the better solutions to generate $X_{t+1}$;
11: **end for**

---

where $x_{r\cdot}$ are randomly chosen solutions and $F1, F2 \in [0, 1]$ are two controllable parameters. The Exponential crossover operator is formulated as:

$$x_i'' = \begin{cases} x_{i,j}', & \text{if } rand_{k:j} < Cr \text{ and } k \le j \le L + k \\ x_{i,j}, & \text{otherwise} \end{cases} , j = 1, \cdots, Dim \tag{12}$$

where $Dim$ is the solution dimension, $L \in \{1, \cdots, Dim\}$ is a random length, $rand \in [0, 1]^{Dim}$ is a random vector, $x_i'$ is the trail solution generated by mutation operator and $Cr \in [0, 1]$ is a controllable parameter. At the beginning, a population $X$ with size 100 is uniformly sampled and evaluated. In each optimization generation, given the parameters $F1, F2, Cr$ from the meta-level agent, algorithm applies DE/current-to-rand/1 mutation and Exponential crossover operator on the population to generate the trail solution population $X_t''$. An comparison is conducted between population $X_t$ and $X_t''$ where the better solutions are selected for the next generation population $X_{t+1}$. Finally the worst solutions are removed from $X_{t+1}$ in the LPSR process.

For $Alg9$ (as shown in Algorithm 2), the population sampled in Halton sampling (Halton, 1960) is divided into four sub-populations. The first sub-population uses GA operators MPX (Holland, 1992) crossover and Polynomial mutation (Dobnikar et al., 1999) accompanying with the Roulette selection (Holland, 1992). MPX crossover is formulated as:

$$x_i' = \begin{cases} x_{r1,j}', & \text{if } rand_j < Cr_1 \\ x_{i,j}', & \text{otherwise} \end{cases} , j = 1, \cdots, Dim \tag{13}$$

where $rand_j \in [0, 1]$ are random numbers, $Cr_1$ is a controllable parameter and $x_{r1}$ is a random solution. The sample method of $x_{r1}$ is also a controllable action $Xr_{mpx}$ which can be uniform sampling or sampling with fitness based ranking. The Polynomial mutation is as follow:

$$x_i'' = \begin{cases} x_i' + ((2u)^{\frac{1}{1+\eta_m}} - 1)(x_i' - lb), \text{if } u \le 0.5; \\ x_i' + (1 - (2 - 2u)^{\frac{1}{1+\eta_m}})(ub - x_i'), \text{if } u > 0.5. \end{cases} \tag{14}$$

where $\eta_m \in \{1, 2, 3\}$ is a controllable parameter, $u \in [0, 1]$ is a random number, $ub$ and $lb$ are the upper and lower bound of the search range.

The second sub-population uses SBX crossover (Deb et al., 1995), Gaussian mutation (Holland, 1992) and Tournament selection Goldberg & Deb (1991):

$$x_i' = 0.5 \cdot [(1 \mp \beta)x_i + (1 \pm \beta)x_{r1}], \text{where } \beta = \begin{cases} (2u)^{\frac{1}{1+\eta_c}} - 1, \text{if } u \le 0.5; \\ (\frac{1}{2-2u})^{\frac{1}{1+\eta_c}}, \text{if } u > 0.5. \end{cases} \tag{15}$$

where $\eta_c \in \{1, 2, 3\}$ is controllable parameter and $u \in [0, 1]$ is random number. Similar to MPX, SBX also uses an action $Xr_{sbx}$ to select parent solutions $x_{r1}$. The Gaussian mutation operator applies Gaussian noise with controllable parameter $\sigma \in [0, 1]$ on the solution:

$$x_i'' = \mathcal{N}(x_i', \sigma \cdot (ub - lb)) \tag{16}$$

The third sub-population is DE/rand/2/exponential (Storn & Price, 1997) where the DE/rand/2 mutation operator is:

$$x_i' = x_{r1} + F1_3(x_{r2} - x_{r3}) + F2_3(x_{r4} - x_{r5}) \tag{17}$$

---

**Algorithm 2** Pseudo code of $Alg9$

---

1: **Input**: Optimization problem $f$, optimization horizon $T$, Meta-level agent $\pi$.
2: **Output**: Optimal solution $x^* = \underset{x \in X}{\arg\min} f(x)$.
3: Initialize 4 sub-populations $\{X_{i,1}\}_{i=1,2,3,4}$ using Halton sampling with sizes $\{200, 100, 100, 100\}$.
4: Evaluate the sub-populations with problem $f$;
5: **for** $t = 1$ **to** $T$ **do**
6:     Receive the 16 action values $a_t$ from the agent $\pi$;
7:     Generate $X_{1,t+1}$ using MPX (Eq. (13)), Polynomial mutation (Eq. (14)) and Roulette selection on $X_{1,t}$;
8:     Generate $X_{2,t+1}$ using SBX (Eq. (15)), Gaussian mutation (Eq. (16)) and Tournament selection on $X_{2,t}$;
9:     Generate $X_{3,t+1}$ using DE/rand/2 mutation (Eq. (17)), Exponential crossover (Eq. (12)) on $X_{3,t}$;
10:     Generate $X_{4,t+1}$ using DE/current-to-best/1 mutation (Eq. (18)), Binomial crossover (Eq. (19)) on $X_{4,t}$;
11:     **for** $i = 1$ **to** 4 **do**
12:         Replace the worst solution in $X_{i,t+1}$ by the best solution in $X_{cm_i,t+1}$
13:     **end for**
14: **end for**

---

where $x_{r.}$ are randomly selected solutions and $F1_3, F2_3 \in [0,1]$ are controllable parameters for the third sub-population. The Exponential crossover formulated as Eq. (12) is used in this sub-population with parameter $Cr_3 \in [0,1]$.

The last sub-population is DE/current-to-best/1/binomial (Storn & Price, 1997). The mutation operator with parameter $F1_4, F2_4 \in [0,1]$ is formulated as:

$$x'_i = x_i + F1_4(x^* - x_i) + F2_4(x_{r1} - x_{r2}) \tag{18}$$

where $x^*$ is the best performing solution in the sub-population. The Binomial crossover uses a similar process as MPX but introduces a randomly selected index $jrand \in \{1, \cdots, Dim\}$ to ensure the difference between the generated solution and the parent solution:

$$x''_i = \begin{cases} x'_{i,j}, & \text{if } rand_j < Cr_4 \text{ or } j = jrand \\ x'_{i,j}, & \text{otherwise} \end{cases} \quad , j = 1, \cdots, Dim \tag{19}$$

where $rand_j$ are random numbers and $Cr_4 \in [0,1]$ is the controllable parameter.

Besides, $Alg9$ conducts the controllable information sharing among the sub-populations where the worst solution in current sub-population $X_{i,g}$ is replaced by the best solution from the target sub-population $X_{cm_i,g}$, $cm_{\{1,2,3,4\}} \in \{1,2,3,4\}$ are four actions indicating the target sub-population.

Given the 16 actions $\{Cr_1, Xr_{mpx}, \eta_m, \eta_c, Xr_{sbx}, \sigma, F1_3, F2_3, Cr_3, F1_4, F2_4, Cr_4, cm_1, cm_2, cm_3, cm_4\}$, $Alg9$ uses these parameters to configure the mutation and crossover operators and applies them on the 4 sub-populations. Then the information sharing is activated for better exploration. Finally, the next generation population is obtained through the population reduction processes.

### D.2 Train-test split of BBOB Problems

As shown in Table 5, the BBOB testsuite (Hansen et al., 2021) contains 24 different optimization problems with diverse characteristics such as unimodal or multi-modal, separable or non-separable, high conditioning or low conditioning. To maximize the problem diversity of the training problem set and hence empower the agent better generalization ability, we choose the most diverse 16 problem instance for training, whose fitness landscapes in 2D scenario are shown in Figure 4. The rest 8 instances are used as testing set whose 2D landscapes are shown in Figure 5. The dimensions of each problem instances in both training and testing set are randomly chosen from $\{5, 10, 20, 50\}$.

### D.3 Baseline Implementation

**RLPSO** (Wu & Wang, 2022) uses two MLP policy networks to configure the algorithm parameters. For each solution in each optimization generation, given the solution and the best so far solution, RLPSO generates the a pair of $\mu$ and $\sigma$ of the target parameter using the two networks respectively.

Table 5: Overview of the BBOB testsuites.

| | Problem | Functions | Dimensions |
|---|---|---|---|
| Separable functions | $f_1$ | Sphere Function | 50 |
| | $f_2$ | Ellipsoidal Function | 5 |
| | $f_3$ | Rastrigin Function | 5 |
| | $f_4$ | Buche-Rastrigin Function | 10 |
| | $f_5$ | Linear Slope | 50 |
| Functions with low or moderate conditioning | $f_6$ | Attractive Sector Function | 5 |
| | $f_7$ | Step Ellipsoidal Function | 20 |
| | $f_8$ | Rosenbrock Function, original | 10 |
| | $f_9$ | Rosenbrock Function, rotated | 10 |
| Functions with high conditioning and unimodal | $f_{10}$ | Ellipsoidal Function | 10 |
| | $f_{11}$ | Discus Function | 5 |
| | $f_{12}$ | Bent Cigar Function | 50 |
| | $f_{13}$ | Sharp Ridge Function | 10 |
| | $f_{14}$ | Different Powers Function | 20 |
| Multi-modal functions with adequate global structure | $f_{15}$ | Rastrigin Function (non-separable counterpart of F3) | 5 |
| | $f_{16}$ | Weierstrass Function | 20 |
| | $f_{17}$ | Schaffers F7 Function | 50 |
| | $f_{18}$ | Schaffers F7 Function, moderately ill-conditioned | 50 |
| | $f_{19}$ | Composite Griewank-Rosenbrock Function F8F2 | 10 |
| Multi-modal functions with weak global structure | $f_{20}$ | Schwefel Function | 20 |
| | $f_{21}$ | Gallagher's Gaussian 101-me Peaks Function | 20 |
| | $f_{22}$ | Gallagher's Gaussian 21-hi Peaks Function | 10 |
| | $f_{23}$ | Katsuura Function | 20 |
| | $f_{24}$ | Lunacek bi-Rastrigin Function | 20 |
| | | Default search range: $[-5, 5]^{Dim}$ | |

Then the parameter value is sampled from $\mathcal{N}(\mu, \sigma)$ and the two policy networks are updated by policy gradient. The original design of using the solution and best solution as network input hinders the generalization ability of the RLPSO policy across problems with different dimensions, therefore in the experiment we replace the network input by the same 9-dimensional state representation as Q-Mamba. To control the algorithms with up to 16 actions in our experiment, we set the output dimension of the two networks to 16 and use the first few values if the number of actions of the algorithm is lower than 16. In summary, for RLPSO baseline we use the MLP with structure ($9 \times 64 \times 32 \times 16$) for both networks and retain their original Policy Gradient training process.

**LDE** (Sun et al., 2021) adopts a Long Short-Term Memory (LSTM) network to integrate the optimization information from previous optimization generations and the fitness of solutions in current population. Then two MLP networks predict the $\mu$ and $\sigma$ for the target parameters of each solution based on the integrated optimization status. REINFORCE is used to update the policy at the end of an optimization trajectory. LDE configure the individual-level parameters for each solution therefore its state representation and action design are related to the population size. To adapt the network to our generated algorithms where population sizes may reduce, we conduct the modification similar to that on RLPSO: we use our 9-dimensional state instead of its original population size related state. The output dimensions of the networks are also set to 16 to perform the population-level parameter configuration. In this paper, we use an one-layer LSTM with input dimension 9 and hidden dimension 32. The MLP network for $\mu$ and $\sigma$ are both ($32 \times 16$).

**GLEET** (Ma et al., 2024b) designs a feature embedding module for feature extraction, a Transformer-based fully informed encoder for information processing amongst individuals and an exploration-exploitation decoder for individual-level parameter configuration in which the encoded individual features are decoded in a Transformer block and generate the individual-wise $\mu$s and $\sigma$s for action sampling. The problem dimension-free state representation and Transformer-based network structure make GLEET compatible to our generated algorithms and problems. Therefore we retain its network designs except the output dimension: a meanpooling is conducted on the decoded features in the exploration-exploitation decoder to transform the individual-level features into a population feature, then 16-dimensional $\mu$ and $\sigma$ are predicted by two MLPs.

**The Decision Transformer** adopts a trajectory-based learning approach, utilizing a Transformer architecture to model the decision-making process from sequential data. It consists primarily of three components: a trajectory embedding module for embedding state-action-return sequences, a Transformer-based decision module for processing sequential information, and a policy decoder for

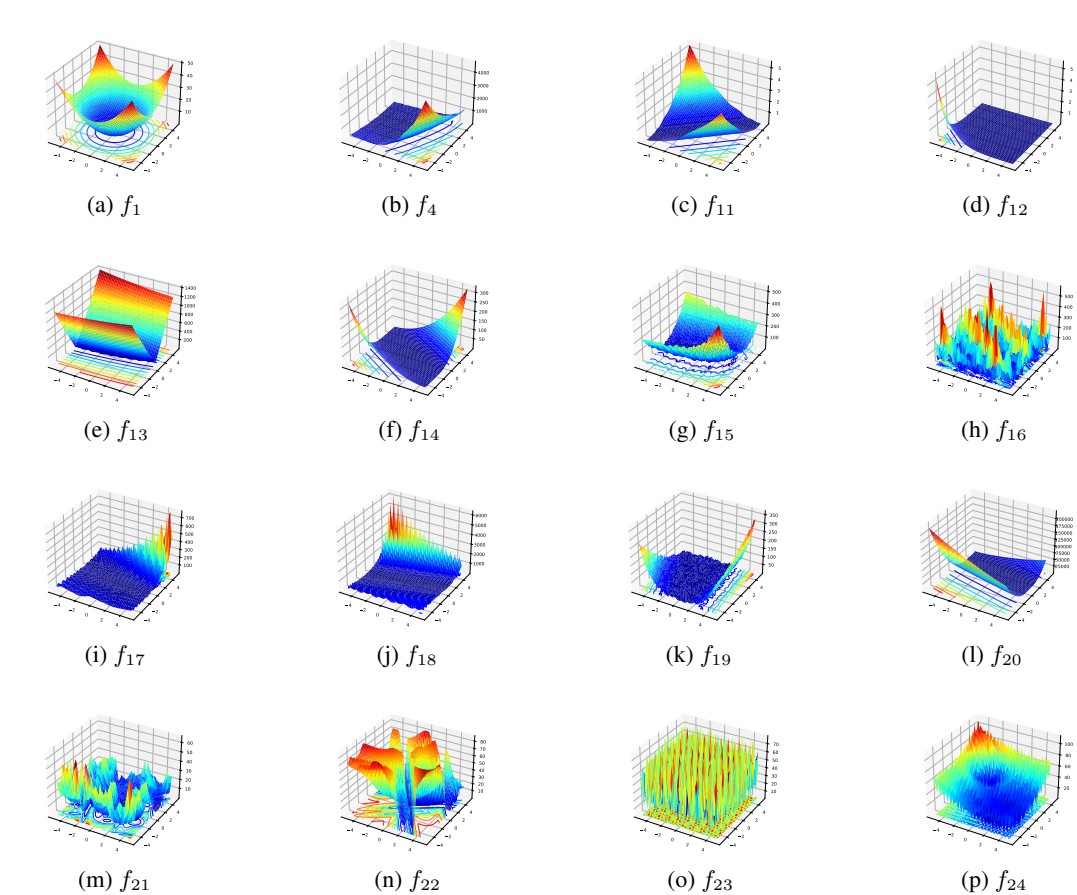

Figure 4: Fitness landscapes of functions in BBOB **train** set when dimension is set to 2.

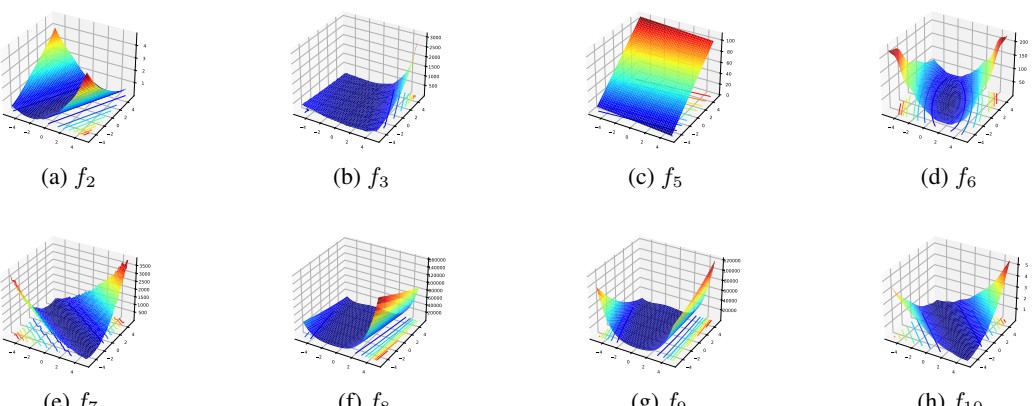

Figure 5: Fitness landscapes of functions in BBOB **test** set when dimension is set to 2.

action generation. The trajectory embedding module encodes states, actions, and returns into token sequences. These tokens are processed through standard Transformer encoder blocks, leveraging self-attention mechanisms to capture long-range dependencies within the trajectory. The encoded sequences are then passed to the policy decoder, which generates predictions for the next action based on the observed past states, actions, and expected returns. The Transformer-based structure enables the Decision Transformer to handle sequences of varying lengths and complex state-action dynam-

ics. In our task, each action space dimension of the original Decision Transformer is treated as an independent token, changing the input sequence format from the original DT's $(R_1, s_1, a_1, R_2, s_2, \ldots)$ to $(R_1, s_1, a_{1_1}, a_{1_2}, \ldots, a_{1_n}, R_2, s_2, \ldots)$ to avoid the exponential growth of the action space.

**Q-Transformer** is a scalable offline reinforcement learning approach that employs a Transformer-based architecture to model Q-functions for multi-task policies. This method discretizes each dimension of the action space, treating each as a separate token, which facilitates auto-regressive Q-learning through effective sequence modelling techniques. By adopting this strategy, Q-Transformer effectively mitigates the exponential growth of the action space, making it well-suited for large-scale offline reinforcement learning tasks. A notable feature of Q-Transformer is its implementation of conservative Q-function regularization, which addresses distributional shifts in conjunction with n-step returns to improve learning efficiency. In our implementation, we utilize a linear action encoder, a single-layer Transformer encoder combined with an MLP as a Q-value head to maintain a compact model size with Q-Mamba.

