# OpenReview forum: "Meta-Black-Box-Optimization through Offline Q-function Learning with Mamba Architecture"
_ICLR.cc/2025/Conference — ICLR 2025 Conference Withdrawn Submission_

### Official Review · Reviewer_uKGW · 2024-10-30

**Soundness:** 3
**Presentation:** 2
**Contribution:** 3
**Rating:** 5
**Confidence:** 3

**Summary:**

This paper proposes a Decomposed Q-function Representation (DQR), enabling sequence modeling-based representation for each component’s Q-function to simplify the learning of an optimal control policy within the entire configuration space of a black-box optimizer. Specifically, using DQR and the Mamba architecture, the paper introduces Q-Leaner, which models each component configuration in the optimizer as a separate time step, auto-regressively predicting the corresponding decomposed Q-function conditioned on the current optimization status and previously selected component configurations, with CQL regularization. Experimental results demonstrate that Q-Mamba achieves competitive optimization performance compared to prior online/offline learning baselines, while requiring at most half the training budget of the online baselines.

**Strengths:**

1. The application of sequence modeling for MetaBBO is intriguing.
2. The paper is well-written.

**Weaknesses:**

1. Including the algorithms for Q-Mamba would enhance the clarity of the optimization pipeline, detailing the input and output to facilitate reader comprehension.

**Questions:**

1. What motivated the choice of Mamba as the sequence modeling architecture? In Table 1, the inference time of Decision Transformer (DT) is comparable to that of Q-Mamba. What primarily drives the performance differences between DT and Q-Mamba—is it due to the additional Q-module? Additionally, how would Q-Mamba compare to QT [1] or QDT [2], which also integrate Q-modules into DT to enhance performance?
2. How does Q-Mamba perform in comparison with other methods in the D4RL environment, a widely used testbed for offline RL methods?
3. Could the authors clarify the problem definition and the necessity of using an offline RL approach in a more accessible way? As I understand, the primary focus of this research is to identify components of the BBO optimization algorithm that enhance generalization.
4. The total training objective is directly defined by the Bellman backup update in Equation 8, differing from the supervised learning objective typically used in sequence modeling for offline RL. How, then, is DT applied in this context? What is the training objective for DT in this setting?

[1] Shengchao Hu, et al. Q-value Regularized Transformer for Offline Reinforcement Learning. 2024 ICML.

[2] Taku Yamagata, et al. Q-learning Decision Transformer: Leveraging Dynamic Programming for Conditional Sequence Modelling in Offline RL. 2023 ICML.

---

### Official Review · Reviewer_6kfW · 2024-11-01

**Soundness:** 3
**Presentation:** 4
**Contribution:** 2
**Rating:** 5
**Confidence:** 3

**Summary:**

Despite the promise of the MetaBBO framework, effective generalization is challenging due to the inefficiency of existing online learning methods. This paper presents Q-Mamba, an innovative offline learning MetaBBO approach.

It features a decomposed Q-function representation to simplify training in extensive joint-action spaces, a Mamba-based Q-Learner for enhanced long-sequence modeling, and CQL regularization for stability. As a result, Q-Mamba performs competitively against current baselines, requires less training budget, and is easily adaptable to new BBO tasks.

Along with architectural contribution, the authors contribute an offline MetaBBO dataset, named E&E dataset, for the first time.

**Strengths:**

- Clear and concise writing style, making the content easy to follow and understand.
- This work makes a well-motivated and reasonable contribution to the optimization domain by addressing practical problems. Introducing an offline RL framework enhances the efficiency of MetaBBO. Additionally, the architectural contribution of Q-mamba further accelerates this efficiency.
- The baselines are thoughtfully designed to offer insightful comparisons within the rich context of MetaBBO. Core architectures for meta-level control policy are generally investigated in both online and offline setups to ensure a comprehensive evaluation.
- As far as I know, this is the first investigation of Decomposed Q Learning with Mamba. Also, this paper’s insights and results can also be generalized to the broader offline RL problem.
- Valuable contribution to the MetaBBO research by providing offline dataset, with a thoughtful design that balances exploration and exploitation by combining high-quality data with randomly generated data.

**Weaknesses:**

- Typo:
   - "very lone sequence" should be "very long sequence" (line 319).
   - Equation 6 (line 311) contains a typo: $E^t_1 \to E^t_i$.

- The approach effectively addresses the practical problem, though the architecture designs are somewhat conventional with limited contributions at the framework level. The experiments yield expected results without any surprising findings.
- The performance of Q-Mamba and the baselines in Table 1 appear quite similar with large variance. So, this might not demonstrate the true efficiency of models. It would be more informative to compare the time (or steps) taken for the model to reach converged performance. Visualizing performance throughout the optimization process, as depicted in Figure 3, could enhance clarity.

**Questions:**

- In the Q-value head, the MLP takes $[s^t, token(a^t_{i-1})]$ along with decision information $\mathbb{O}^t_i$. Can you provide any intuition or empirical evidence for this design choice?
- Can you provide more details about the mamba_block, such as the number of layers used? If only one block of Mamba is used as I understood from the paper, is it fair to compare it with Q-Transformer in terms of model capability? I wonder if we reduce the layer in Q-transformer the performance drops so significantly.
- Why does the online Q-Mamba approach take significantly longer to train compared to the Transformer-based online approach? Is this due to the introduction of the decomposed Q function? While the massive training time for Q-Transformer is understandable, what is the primary reason for the difference in training time here?

---

### Official Review · Reviewer_VHiT · 2024-11-04

**Soundness:** 3
**Presentation:** 2
**Contribution:** 2
**Rating:** 6
**Confidence:** 4

**Summary:**

The paper introduces Q-Mamba, an offline learning framework for Meta-Black-Box Optimization (MetaBBO) designed to improve both the optimization performance and training efficiency of existing online MetaBBO methods. Q-Mamba employs a Decomposed Q-function Representation (DQR) to address the challenge of controlling a low-level optimizer’s multiple configurable components, breaking down the complex joint-action space into sequential Q-functions for each component. The framework uses a Mamba-based neural network Q-Learner to model these sequential Q-functions, enhancing long-sequence modeling for improved policy learning. By leveraging offline trajectory data from both exploration and exploitation, Q-Mamba aims to circumvent issues associated with online learning, such as high training costs and inefficiency. Additionally, Q-Mamba integrates Conservative Q-Learning (CQL) regularization to address distributional shifts during offline training. Experimental results indicate that Q-Mamba performs competitively, often surpassing both online and offline baselines in efficiency and effectiveness on a range of black-box optimization tasks.

**Strengths:**

- The paper offers a novel synthesis of several established techniques: it combines the decomposition of action dimensions in the Q-function to increase efficiency, employs the Mamba architecture for improved handling of long sequences, models meta-black-box optimization as a sequential decision-making problem addressed through RL, and utilizes offline reinforcement learning to further enhance efficiency.

- The OOD generalization experiments in Section 5.3 demonstrate remarkable performance by Q-Mamba. Notably, the model, meta-trained on synthetic problems, effectively generalizes to more complex MuJoCo continuous control tasks.

- The paper's presentation is clear and well-motivated, with empirical evaluations that convincingly support the authors’ claims.

**Weaknesses:**

- The concrete problem setup and formal definition of MetaBBO appear only in Section 4.1, making it challenging for readers to fully comprehend the framework in earlier sections. Introducing the MetaBBO problem definition earlier would facilitate a quicker understanding of the framework's main contributions.

- Q-Mamba employs a decomposed Q-function, where each action dimension’s value is estimated auto-regressively. While this approach is effective for handling large joint-action spaces, the paper does not analyze how the ordering of action dimensions might affect the performance. A systematic evaluation of the impact of action dimension ordering would add valuable insights to the empirical results.

- Q-Mamba uses the relative improvement in objective values as its reward, which assumes knowledge of the optimal objective value --- a condition that may not always be practical in real-world applications. The paper would benefit from discussing alternative reward shaping methods and their potential impact on learning outcomes.

- In Section 5.2, the online variant of Q-Mamba underperforms the offline version, which the authors attribute to the advantage of a diverse offline dataset collected from other online methods. However, this result suggests that Q-Mamba may lack effective exploration capabilities during online training, an aspect that could be further analyzed or addressed.

**Questions:**

__[Questions and Comments]__

* Typo: in Line 143 on page 3, shouldn't "improving" be used instead of "reducing" in the given context?

* Notation for Q-functions: why do the authors denote Q-functions as if they are probability distributions? For instance, in Equation 1, the notation used is $Q(a_{1:K}^t|s^t)$. Wouldn’t it be clearer to write this as $Q(s^t, a_{1:K}^t)$? If there is a specific reason for adopting the conditional notation, an explanation would be helpful.

* Reward signal based on relative improvement: as noted in the weaknesses, why did the authors choose to use relative improvement in objective values, rather than, e.g., an absolute value, as the reward signal? This approach assumes access to an optimization oracle. Have the authors tried alternative reward types? Do they anticipate that the performance of Q-Mamba might degrade if optimal objective values are not incorporated?

* Impact of action dimension ordering: also noted in the weaknesses, the paper would benefit from a discussion on the impact of action dimension ordering on performance. How sensitive is Q-Mamba to this ordering, and do the authors have any insights on how different orderings might affect outcomes?

* CQL regularization: in CQL, the values of OOD actions are minimized rather than regularized to zero. Is there a reason why the authors adopt a zero-regularization scheme in Equation 8? Additionally, if my understanding is correct, the relevant discussion in lines 358-362 should be updated to show how this regularization term differs from what's in CQL.

* Exploration capability: why might Q-Mamba exhibit weaker exploration capabilities than other online baselines? Given that Q-Mamba underperforms its offline variant, it would be interesting to understand what limitations might be affecting its exploration during online training.

* In Table 1, the confidence intervals of RLPSO and Q-Mamba largely overlap. However, the authors state that "Q-Mamba significantly outperforms the online baselines RLPSO, LDP, and GLEET, which control...". This statement could be misleading given the overlapping confidence intervals and would benefit from being rephrased.

* Details of experiments in Section 5.3: can the authors provide a detailed description of the experiments conducted in Section 5.3 and consider adding this information to the appendix?

---

### Official Review · Reviewer_zkyX · 2024-11-04

**Soundness:** 2
**Presentation:** 1
**Contribution:** 2
**Rating:** 3
**Confidence:** 3

**Summary:**

This paper introduces Q-Mamba, a novel offline reinforcement learning framework for Meta-Black-Box Optimization (MetaBBO). The main idea is to leverage Mamba's long-sequence modeling advantage to represent the Q function in MetaBBO. To do this, they train the Q-Mamba offline, which they claim is more efficient than the online counterpart. The authors constructed a dataset to evaluate the performance of the proposed framework.

**Strengths:**

The paper is clearly written. Several strengths of this work:

The offline learning strategy is shown to significantly improve training efficiency compared to online baselines, a major advantage in computationally expensive MetaBBO tasks.

The paper presents extensive benchmarking results demonstrating that Q-Mamba achieves competitive or superior performance compared to various online and offline baselines, including on unseen tasks (zero-shot generalization).

Ablation studies provide evidence supporting the effectiveness of the key design choices (decomposed Q-function, Mamba architecture, CQL regularization, E&E dataset).

**Weaknesses:**

1. The paper only briefly mentions the following two reasons for using Mamba in Section 4.2:

●The sequence length of MetaBBO tasks involves thousands of decision steps. Transformer suffers from inefficiency and performance issues when dealing with very long sequences, while Mamba addresses this problem through data-dependent embedding and hardware-aware design.

●
Mamba allows for the selective extraction of essential information and filtering out of irrelevant noise based on the input sequence, which effectively enhances sequence-to-sequence learning.

However, the paper lacks a detailed analysis of the following:

●
How Mamba improves the efficiency and performance of extremely long sequence modeling through data-dependent embedding and hardware-aware design.

●
The specific performance improvements and efficiency advantages of Mamba in processing extremely long Q-function sequences compared to Transformer.

●
How Mamba selectively extracts important information and filters out irrelevant noise based on the input sequence, and how this mechanism enhances sequence-to-sequence learning.

2. The performance heavily relies on the quality and composition of the E&E dataset. Further investigation into the sensitivity to dataset variations would strengthen the claims.

3. While zero-shot generalization is demonstrated, the extent of generalizability to significantly different optimization problems beyond those tested remains unclear. More diverse and challenging test scenarios are needed.

4. Limitation on the Number of Action Bins: The number of action bins (M) used in Q-Mamba cannot be too large, which may be a limitation for some optimizers that require fine-grained control.

**Questions:**

1. The paper mentions a performance degradation of Q-Mamba in the online learning setting. Does this mean that offline data is always more effective than online data in MetaBBO?

2. For those optimizers with continuous action spaces, how does Q-Mamba deal with the information loss caused by action discretization?

---

### Note · Authors · 2025-01-19

**Comment:**

N/A

**Withdrawal Confirmation:**

I have read and agree with the venue's withdrawal policy on behalf of myself and my co-authors.